# A glucose meter interface for point-of-care gene circuit-based diagnostics

Evan Amalfitano[1,16], Margot Karlikow[1,16], Masoud Norouzi[1,16], Katariina Jaenes[1], Seray Cicek[1], Fahim Masum[1], Peivand Sadat Mousavi [1], Yuxiu Guo[1], Laura Tang [1], Andrew Sydor [2], Duo Ma[3], Joel D. Pearson[4,5,6], Daniel Trcka[4], Mathieu Pinette[7], Aruna Ambagala[7], Shawn Babiuk [7], Bradley Pickering[7,8,9], Jeff Wrana[4,10], Rod Bremner [4,5,6], Tony Mazzulli[6,11], David Sinton[12], John H. Brumell [2,10,13,14], Alexander A. Green [3,15] & Keith Pardee [1,12✉]

Recent advances in cell-free synthetic biology have given rise to gene circuit-based sensors with the potential to provide decentralized and low-cost molecular diagnostics. However, it remains a challenge to deliver this sensing capacity into the hands of users in a practical manner. Here, we leverage the glucose meter, one of the most widely available point-of-care sensing devices, to serve as a universal reader for these decentralized diagnostics. We describe a molecular translator that can convert the activation of conventional gene circuit-based sensors into a glucose output that can be read by off-the-shelf glucose meters. We show the development of new glucogenic reporter systems, multiplexed reporter outputs and detection of nucleic acid targets down to the low attomolar range. Using this glucose-meter interface, we demonstrate the detection of a small-molecule analyte; sample-to-result diagnostics for typhoid, paratyphoid A/B; and show the potential for pandemic response with nucleic acid sensors for SARS-CoV-2.

[1] Leslie Dan Faculty of Pharmacy, University of Toronto, Toronto, ON M5S 3M2, Canada. [2] Program in Cell Biology, Hospital for Sick Children, Peter Gilgan Center for Research and Learning, Toronto, ON M5G 0A4, Canada. [3] Biodesign Center for Molecular Design and Biomimetics, The Biodesign Institute and the School of Molecular Sciences, Arizona State University, AZ 85287, USA. [4] Lunenfeld Tanenbaum Research Institute, Mt Sinai Hospital, Sinai Health System, Toronto, M5G 1X5 ON, Canada. [5] Department of Ophthalmology and Vision Science, University of Toronto, Toronto, M5T 3A9 ON, Canada. [6] Department of Laboratory Medicine and Pathobiology, University of Toronto, Toronto, M5S 1A8 ON, Canada. [7] Canadian Food Inspection Agency, National Centre for Foreign Animal Disease, Winnipeg, R3E 3M4 MB, Canada. [8] Department of Medical Microbiology and Infectious Diseases, Faculty of Medicine, University of Manitoba, Winnipeg, R3E 0J9 MB, Canada. [9] Iowa State University, College of Veterinary Medicine, Department of Veterinary Microbiology and Preventive Medicine, Ames, IA 50011, USA. [10] Department of Molecular Genetics, University of Toronto, Toronto, M5S 1A8 ON, Canada. [11] Department of Microbiology, Sinai Health System/University Health Network, Toronto, M5G 1X5 ON, Canada. [12] Department of Mechanical and Industrial Engineering, University of Toronto, Toronto, M5S 3G8 ON, Canada. [13] Institute of Medical Science, University of Toronto, Toronto, M5S 1A8 ON, Canada. [14] SickKids IBD Centre, Hospital for Sick Children, Toronto, M5G 1X8 ON, Canada. [15] Department of Biomedical Engineering, Boston University, Boston, MA 02215, USA. [16]These authors contributed equally: Evan Amalfitano, Margot Karlikow, Masoud Norouzi. ✉email: keith.pardee@utoronto.ca

The field of synthetic biology, now on the verge of its third decade, is emerging from its formative tool-building phase into one that is application focused[1–3]. This trend toward practical implementation is setting the stage for biology to address unmet needs in health and food security[4–9], green manufacturing and energy[10], amongst many others[11,12]. Work to date in the field has largely relied on encoding functions into cells, which can take months or years to reach a final product[13]. This has led to efforts to move synthetic biology out of the cell using cell-free protein expression systems which vastly accelerates the design-test cycle[14–16] and enables these biotechnologies to be housed in a biosafe format that can be freeze-dried for distribution and use without refrigeration[4,9,17–19]. Using the cell-free enzymatic machinery of transcription and translation, these systems allow for gene circuits to operate as they do inside living cells[17,20].

Efforts to develop cell-free technologies have been particularly successful for point-of-care (PoC) diagnostics, environmental sensing[4,17,21–24], on-demand manufacturing of protein-based therapeutics[9,18,19,25] and education[26,27]. One of the most exciting aspects of these efforts is the potential to improve access to health care. This is especially relevant for PoC diagnostics where the principles of rational design from synthetic biology are enabling the rapid and low-cost development of sensors for pathogens and small molecules. These advances are bringing real-time surveillance of outbreaks and low-cost tools for accessible global health closer to reality. As demonstrated by the SARS-CoV-2 outbreak, access to diagnostics is critical and shortages can be a significant bottleneck to containment, leading to restrictions in movement and normal life[28]. While gene circuit-based sensors provide a promising solution, they need to be paired with companion technologies to be used outside of the laboratory.

Recognizing the importance of this challenge, we identified the blood glucose meter as a potential solution. As one of the most widely used PoC monitoring devices, it has significantly improved the lives of millions of people with diabetes by enabling the portable quantification, and therefore personal management, of blood sugar levels. These devices are based on a glucose-dependent electrochemical signal generated by an enzyme, such as glucose oxidase, in their test strips[29]. The widespread adoption of glucose monitors has resulted in a global network of device manufacturing and consumable distribution, as well as broad acceptance by patients and clinicians. Inspired by this model, we aimed to leverage this established diagnostic infrastructure as a universal interface for emerging diagnostic toolsets from the field of synthetic biology. Previous work has shown that glucose meters can be used for the detection of analytes other than glucose, particularly through the release of a pre-existing, DNA-invertase conjugate from magnetics beads. However, to date[30–35] the potential of a universal glucose-based output platform for gene circuit-based sensors and their use in real-world applications has yet to be demonstrated. Importantly, here we report the de novo design and broad application of gene circuit-based sensors capable of the in situ expression of glucogenic reporter enzymes in response to target analytes.

Below we describe the development of a gene circuit-based system that translates the presence of a target analyte into a signal that can be easily measured by a glucose meter. This is achieved by replacing conventional reporter proteins for gene circuit-based sensors (e.g. green fluorescent protein) with enzymes that convert inert glucose-containing precursors into glucose monomers that can be detected by an off-the-shelf glucose meter. The development of this platform begins with identifying glucogenic enzymes that are compatible with cell-free systems and that can serve as novel reporter proteins. Using a typical commercial glucose meter, we screened candidate enzymes and found that the expression of select enzymes can generate glucose in as little as 60 min in coupled protein synthesis-glucose generation reactions. We also demonstrate an enzyme-mediated method with the potential for pre-clearing endogenous glucose from samples (e.g. blood). Next, we use the substrate-specific nature of reporter enzymes, along with other strategies, to allow simple sensor multiplexing in cell-free reactions. With the glucose-mediated interface established, we then demonstrate the successful operation of gene circuit-based sensors for small molecules and synthetic RNAs. To show the potential of this interface for global health applications, we utilize the glucose meter to detect RNA sequences for typhoid, paratyphoid A and B, and related drug resistance genes. Using the related RNAs, we describe robust signal within 60 minutes, sensor multiplexing and detection of *Salmonella typhi* itself at clinically relevant concentrations. Finally, to demonstrate the potential of this platform for response to public health emergencies, we present the development and validation of sensors for the SARS-CoV-2 virus, the causative agent of the current global COVID-19 pandemic, and demonstrate detection from clinical samples.

## Results

**Platform development**. We first assessed the feasibility of glucose detection in the biochemical context of a cell-free system (CFS) (Fig. 1a). We chose a recombinant CFS (PURExpress, NEB) that is comprised of purified proteins and does not contain enzymes from glucose metabolism that would interfere with glucose generation in our system. By simply doping glucose into the CFS we tested whether an off-the-shelf glucose meter with test strips could detect the presence of the sugar. Our initial efforts were confounded by a high degree of read variability, which, after buffer optimization, was resolved by the simple addition of 0.0125 % Tween-20, a non-ionic detergent. The detergent may serve to ensure that the CFS efficiently wicks into the capillary channel of the test strips.

With the concept of measuring glucose from CFS established in principle, our next step was to develop reporter enzyme systems with the capacity to generate glucose in the presence of substrate. With these in-hand, gene circuit-based sensors can be designed to generate glucose in response to activation (Fig. 1b). We began by screening the de novo expression and catalytic activity of 40 enzymes under the buffer conditions of the CFS. Leveraging the capacity of the CFS for rapid prototyping, we tested the expression and corresponding substrate catalysis by the enzymes in an overnight incubation. Using the glucose meter to detect glucose output, we found three enzymes that yielded significant glucose level increases in CFS (Fig. 1c): trehalase (*C. japonicus*, *tre37a*), lactase (*E. coli*, *lacZ*) and phosphatase (*E. coli*, *ybiV*), using their respective substrates trehalose, lactose, and glucose-6-phosphate. These results were corroborated using a separate colorimetric assay for detection, based on the conversion of nicotinamide adenine dinucleotide (NAD) to NADH by glucose dehydrogenase (GDH) in the presence of glucose (Fig. 1d).

To tackle the potential issue of endogenous blood glucose background in patient samples, we used the enzyme glucose dehydrogenase (*B. subtilis*, variant E170K/Q252L[36]). This enzyme from the oxidoreductase family converts D-glucose into D-glucono-1,5-lactone, which is inert to the glucose meter. For every molecule of glucose catabolized, this enzyme requires a molecule of NAD. We hypothesized that this 1:1 stoichiometry between substrate and cofactor might allow us to selectively clear fixed amounts of glucose from incoming samples by adding equimolar amounts of the NAD co-factor. To test the concept, we added glucose to the CFS and evaluated the effect of expression of glucose dehydrogenase in the presence of various concentrations

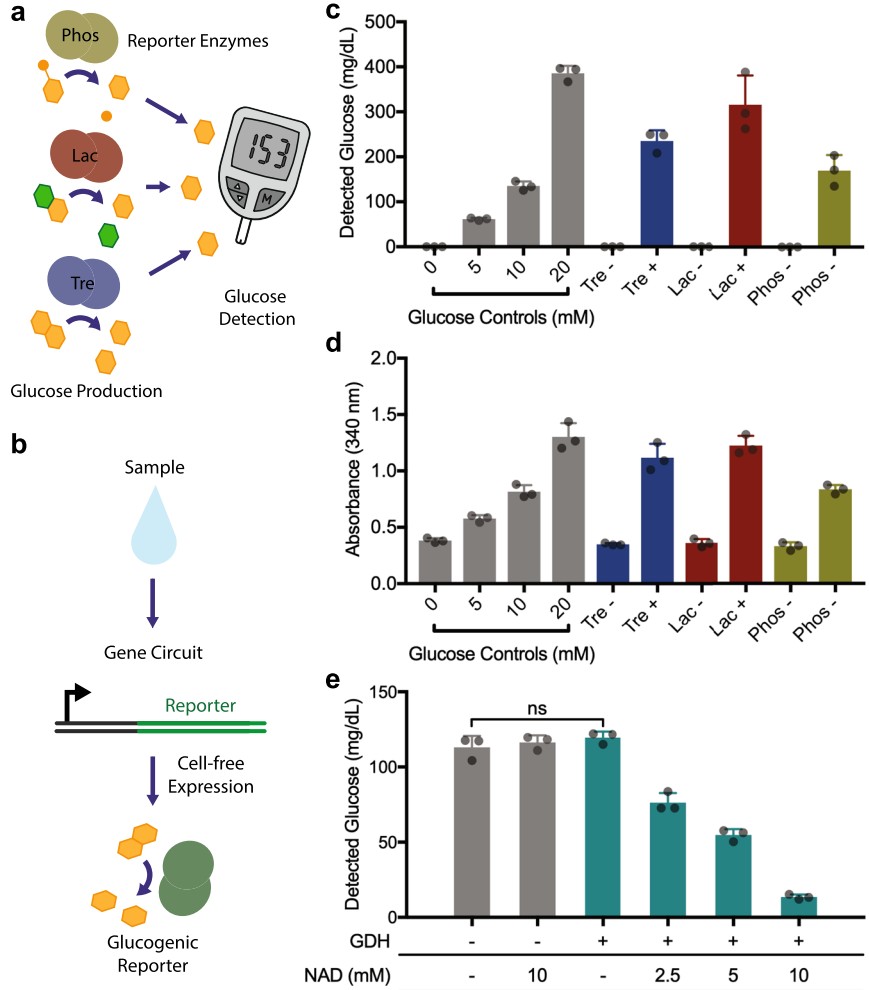

**Fig. 1 Development of a glucose meter interface for cell-free, gene circuit-based sensors. a** Concept schematic of glucose generation by reporter enzymes upon the activation of gene circuit sensors. **b** Expression of different reporter enzymes can be used to generate glucose using their respective substrates. **c** Using a glucose meter for measurement, glucose is generated from the expression of three enzymes: trehalase (Tre), lactase (Lac), phosphatase (Phos). 5 ng/μL plasmid DNA used for expression, reactions incubated for 16 h. Control reactions (-) were identical, but lacked DNA encoding an enzyme. **d** Cell-free protein expression reactions were also analyzed for glucose production using a colorimetric GDH-NAD absorbance-based assay. **e** Using the expression of GDH and a titration of NAD concentration, glucose can be selectively removed from samples. Negative controls (−), without GDH and/or NAD, show no background glucose degradation. GDH DNA added to 10 ng/μL. All data presented are the mean of $N = 3$ independent experiments (as indicated by dot plots) ±SD. Statistical analysis: one-way ANOVA, ns = not significant. Source data are provided as a Source Data file.

of NAD. After an overnight incubation at 37 °C, glucose meter measurements showed that glucose concentration reduction was dependent on glucose dehydrogenase and proportional to supplemented NAD concentration (Fig. 1e). These results highlight that NAD availability can be used effectively to tune glucose dehydrogenase-mediated glucose clearance.

Toehold switches are a class of programmable riboregulators that can be used to control reporter protein expression in a sequence-specific manner, allowing them to serve as RNA sensors[37]. We previously used toehold switches for the recognition of pathogen sequences using colorimetric protein reporters for optical detection[4]. Here, to produce a glucose output from gene circuit-based sensors, we placed the sequence for the trehalase reporter enzyme downstream of a rationally designed toehold switch-based RNA sensor[37]. DNA encoding this toehold switch-based sensor was added to CFS containing trehalose (20 mM) and incubation was limited to one hour to simulate practical applications. At the shorter one hour incubation, CFS reactions contribute a non-specific background signal that is not present in overnight reactions (Fig. 2a). The use of control

reactions containing the sensor alone resolves this potential challenge (see below) by making the detection of target sequences unambiguous.

Glucose meters provide a quantitative measurement of glucose concentration from 10 to 600 mg/dL which introduces the possibility of creating high and low glucose reporter levels to enable multiplexed output. With this in mind, we evaluated the potential for tuning the level of glucose output from the sensors, using synthetic sensor A and its respective target (Supplementary data sets 2-3)[37]. We started by testing the effect of varying the concentration of the substrate trehalose and found that substrate concentration had a significant impact on the glucose signal generated, with higher concentrations providing greater signal-to-noise ratios and stronger signals (Fig. 2a). We also found that the glucose output level was responsive to the concentration of template DNA for the toehold switches (Fig. 2b). Looking to further control glucose production, we investigated the potential of different reporter enzymes to generate variable glucose output levels. When DNA coding for two reporter enzymes (lactase or trehalase) with two different upstream toehold switches was

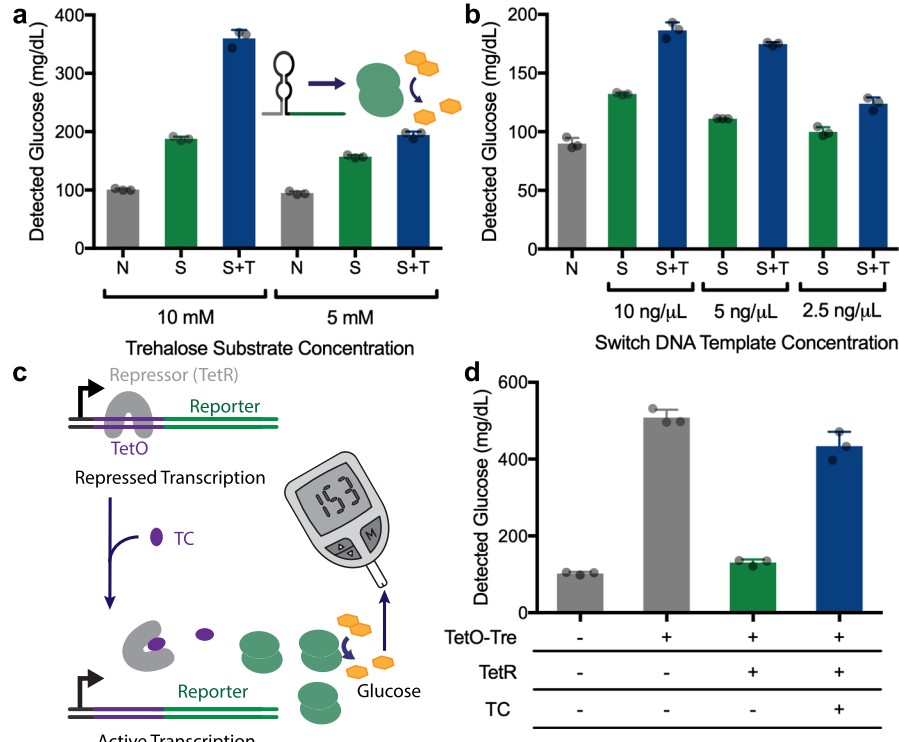

**Fig. 2 Demonstration of gene circuit-based sensors linked to a glucose output. a** Effect of trehalose substrate concentration on glucose output levels in a blank CFS reaction (N, negative control), or reactions with toehold switch-regulated glucose production in the absence (S, switch only) or presence (S + T, switch and trigger) of the corresponding trigger RNA. Switch and trigger added to 10 ng/μL DNA. $N = 3$ biological replicates, error bars represent mean ± SD. **b** Effect of different switch DNA template concentrations on glucose output in the absence (S, switch only) or presence (S + T, switch and trigger) of the corresponding trigger, shown with a CFS without any added DNA (N, negative control) for background comparison. Trehalose present at 5 mM. **c** Schematic of an alternative gene circuit for small molecule detection, based on the TetO/TetR inducible system. The repressor element regulates trehalase expression in response to tetracycline. **d** Demonstration of the system shown in **c** in practice. As seen in the control reactions, TetO-Tre expression results in high glucose production in the absence of both TetR and tetracycline (TC). With the addition of the TetR, glucose production is reduced to near background levels. The addition of both TetR and tetracycline, however, results in a significant increase in glucose output levels. All data presented are the mean of $N = 3$ independent experiments (as indicated by dot plots) ±SD Source data are provided as a Source Data file.

added to the same CFS reaction, different glucose levels were produced depending on which trigger RNA was present in the reaction (Fig. S1a).

Using the rapid (1 h) and high dynamic range of the trehalase reporter, we then took advantage of the gene circuits themselves to modulate glucose production. As we have recently reported[38], toehold switch sequences can influence the expression level of reporter enzymes to provide an additional layer of control in the design of cell-free sensors. By combining a toehold switch with a low expression output (Tre A) and another with a high expression output (Tre B), we show that the sequence-specific detection of the respective RNAs can be distinguished using a common enzyme/substrate pair. Activation of the switches with an equal concentration of RNA triggers A and B (5 nM) results in distinct low and high glucose outputs, respectively (Supplementary Fig. S1b). Importantly, when both target RNA sequences are present, the combined glucose production itself can also be clearly distinguished from the presence of either RNA inputs alone (Supplementary Fig. S1b).

**Glucose meter-mediated small molecule detection**. With the development of the underlying glucose meter-based interface complete, we next focused on demonstrating the potential of this tool for the gene circuit-mediated detection of a small molecule other than glucose. To create a sensor for the antibiotic tetracycline (TC) we adapted a bacterial regulatory gene circuit for antibiotic resistance. In wild-type systems, the transcription of the

tetracycline efflux pump (TetA) is regulated through the TetA operator (TetO) sequence, which is repressed by the binding of a repressor protein (TetR) in the absence of tetracycline. When tetracycline is present, TetR dissociates from TetO enabling the expression of TetA[39]. We recapitulated this circuit in vitro by placing the TetO regulator between a T7 promoter and the trehalase reporter enzyme to create a glucose-generating sensor that is responsive to tetracycline (Fig. 2c). In the absence of TetR, reactions generate a strong glucose signal in comparison to the negative control, but when the TetR protein is added to the control sensor circuit, glucose production is significantly reduced (Fig. 2d). Finally, we show that in the presence of both TetR and tetracycline (TC), high glucose measurements are re-established, demonstrating the ability of the glucose meter to effectively sense the presence of this small molecule.

**Proof-of-concept diagnostic sensors**

*Antibiotic resistance gene sensors.* To test the potential of the glucose meter for PoC molecular diagnostics, we used toehold switches designed for the detection the genes responsible for ampicillin (AmpR) and spectinomycin (SptR) resistance[17] to regulate trehalase reporter expression. DNA encoding the respective switches was then tested in the presence or absence of the RNA target in CFS containing trehalose. Both sensors yielded a strong glucose response in the presence of synthetic target RNA sequences (Supplementary Fig. S2).

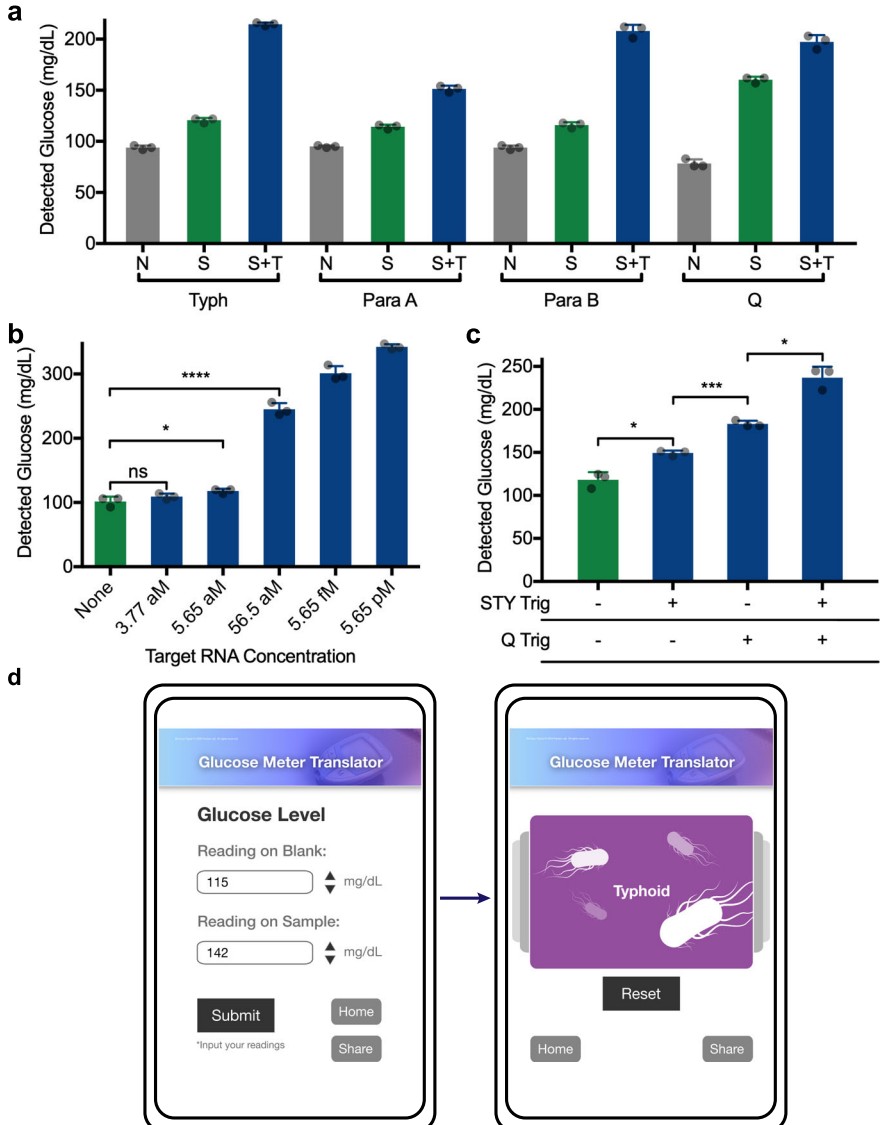

**Fig. 3 Demonstration of typhoid- and paratyphoid-specific toehold switches. a** Glucose output from the detection of RNA sequences related to typhoid (Typh), paratyphoid A (Para A), paratyphoid B (Para B), and fluoroquinolone resistance (Q). CFS background (N, no switch/trigger), switch alone (S, no trigger), and switch with target (S + T) are shown for each experiment. Typh, Para A, Para B switches at 312.5 pg/µL and targets at 50 nM. Q switch at 1.25 ng/µL and target at 100 nM. For each set 3 technical replicates are shown, representative of 3 independent experiments. **b** NASBA amplification of STY RNA at the given initial concentrations followed by glucose generation reaction. Typh switch at 1.25 ng/µL. *$p = 0.0460$; ****$p < 0.0001$. 3 technical replicates are shown, representative of 3 independent experiments. **c** RNA target(s) corresponding to typhoid ("STY") and fluoroquinolone resistance ("Q") were first amplified in a combined NASBA reaction, followed by multiplexed glucose reactions containing trehalase switches for both targets. Typh switch at 1.25 ng/µL, Q at 2.5 ng/µL. *: STY Trig vs. None $p = 0.0180$, STY Trig + Q Trig vs. Q Trig $p = 0.0128$; ***$p = 0.0003$. **d** Web-based interface for automated interpretation of the multiplexed diagnostic data presented in **c**. The user inputs the glucose measurements for a negative control and their experimental sample. Following submission, the website compares the data with known results to provide an interpretation of results (here typhoid positive). ns: not significantly different. All data are presented as mean values ± SD. Source data are provided as a Source Data file.

With the potential for molecular diagnostics established, we set out to develop a panel of tests for typhoid, paratyphoid A and B and the associated fluoroquinolone antibiotic resistance gene QnrS[40]. These infectious diseases are common in global regions where access to clean water is limited. It is estimated that these diseases, collectively referred to as enteric fever, annually infect over 14 million people, causing >135,000 deaths[41]. Despite the widespread impact on health, it remains challenging to effectively diagnose and track these infections due to the lack of low-cost, point-of-care diagnostics[42].

Twelve toehold switches for each of the four target genes (corresponding to typhoid[43], paratyphoid A and B[44], and fluoroquinolone resistance[40]) were computationally designed and placed upstream of the trehalase reporter gene. Adapting the colorimetric GDH-based glucose assay described above (Fig. 1d), we rapidly screened for switch performance using linear DNA (50 nM). Toehold switches with a significant glucose signal output were identified for each target gene (Supplementary Fig. S3) and the top-performing switches were then validated using the glucose meter detection system (Fig. 3a).

We then set out to evaluate the detection threshold of the glucose meter interface when paired with an upstream isothermal amplification step. Previous work has shown that the addition of an amplification step can improve detection sensitivity by orders

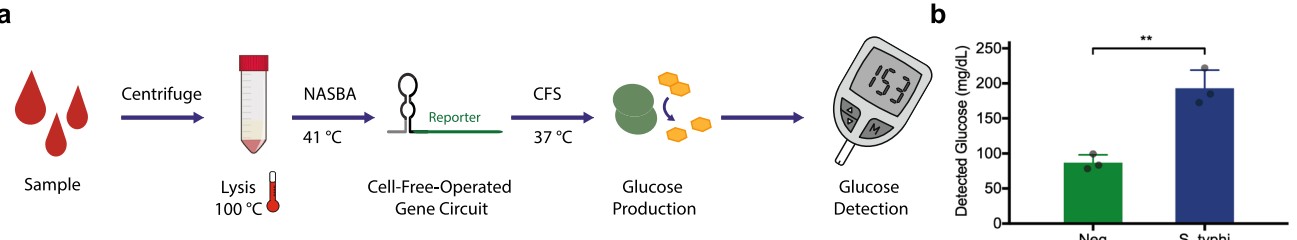

**Fig. 4 Detection of S. typhi using the STY-specific toehold switch. a** Schematic of the workflow used for detection of S. typhi from serum-containing mock samples (7% FBS). **b** Detection of S. typhi cells using this workflow from the serum-containing input with $10^3$ CFU/mL. Typh switch at 1.25 ng/μL. **$**p = 0.0097$. Data presented is the mean of $N = 3$ independent experiments (as indicated by dot plots) ±SD Source data are provided as a Source Data file.

of magnitude, allowing engineered molecular sensors to detect target nucleic acids well within clinically relevant concentrations[4]. For target amplification we chose Nucleic Acid Sequence-Based Amplification (NASBA), which is a primer-directed method that yields significant amplification and a second sequence-specific check point for the diagnostic[45,46]. Using NASBA (1 h) followed by a reaction containing the typhoid-specific toehold switch (targeting the STY1607 gene), the glucose meter interface achieved detection of target RNAs down to the low attomolar range (Fig. 3b). This positive detection of STY RNA from 56 aM samples (1 μL input) represents an initial input of approximately 34 total RNA copies. Similar results were found for paratyphoid A and B, and QnrS targets, with detection thresholds in the low attomolar range (Supplementary Fig. S4). Orthogonality of these sensors was also tested using the glucose meter interface and, as expected, showed specific activation (Supplementary Fig. S5).

We next sought to demonstrate the potential of the glucose meter interface for simple multiplexed diagnostics using the typhoid (STY) and QnrS (Q) sensors. As above, we used NASBA isothermal amplification prior to toehold switch reactions to determine if we could detect each RNA target alone or in combination from low starting concentrations (500 copies in 1 μL). We took advantage of differential glucose outputs from the two sensors to distinguish between detection events. NASBA and toehold switch reactions were performed and using the glucose meter to monitor outputs, we found that STY and Q RNAs yield significantly distinct glucose signals (Fig. 3c). Similarly, when STY and Q RNA inputs were combined (500 copies each), the sensor platform generated a third and unique glucose concentration that allowed for distinction from the individual RNAs. Thus, we showed that the glucose meter interface has the potential for multiplexed detection from input RNAs in the attomolar range. This demonstrates a possible real-world application—a single test capable of independently detecting the presence of both typhoid and a relevant antibiotic resistance gene.

To bring this multiplexed detection capability one step further, we wanted to provide health care professionals with an unambiguous interpretation of the results to provide users with the appropriate course of action. To meet this challenge, we developed a proof-of-concept online tool for interpreting the glucose values (https://www.pardeelab.org/gm.html). Here, using a smartphone or computer, users enter the measured glucose concentration values (mg/dL) for control and test reactions (Fig. 3d, left). These values are compared to look-up tables associated with the tests, and definitive results are displayed (Fig. 3d, right).

Moving the potential of this sensor platform into the hands of users in de-centralized clinics poses challenges, including how to reliably lyse the cell wall of these gram-negative bacteria using simple protocols and equipment. We therefore developed accessible protocols for the detection of endogenous RNA from

whole-cell *S. typhi*. For this we developed a workflow that combines heat and detergent treatments of *S. typhi*, which is compatible with NASBA and subsequent CFS detection of target RNA (Fig. 4a, b)[47]. Using the glucose meter to monitor reaction outputs, we found that the sensor platform could clearly detect target STY RNA from serum-containing mock samples with $10^3$ CFU/mL *S. typhi*. This corresponds to levels found in patients who tested positive for typhoid infections using the most common detection protocol, lab-based blood culture, which has been quantified to include samples in the range of 1000 – 43,500 DNA copies/mL[48]. With lab-based method such as blood culture and PCR difficult to implement in low-resource settings, we envision the glucose meter-based interface providing a more rapid and point-of-need method of typhoid detection[42,48].

The glucose meter interface also holds great potential in providing de-centralized molecular diagnostic capacity to front-line responders in public health emergencies. With the current coronavirus (SARS-CoV-2) outbreak[49,50], there is an urgent need for low-cost, PoC molecular diagnostics that can be distributed beyond centralized facilities. With this in mind, and to demonstrate the adaptable capability of the glucose meter platform for the detection of novel and emerging pathogens, we designed and assembled 24 toehold switches specific to six target regions in the SARS-CoV-2 genome (ORF1b, RdRP, E, and three locations within the N gene: N1, N2, N3). Using a lactase reporter combined with the colorimetric substrate chlorophenol red-beta-D-galactopyranoside (CPRG), we screened all 24 toehold switches for response to the corresponding synthetic trigger RNAs and found 6 high-performing candidates (Supplementary Fig. S6, supplementary dataset 2). Top-performing switches were then tested for trigger-mediated glucose output production using a glucose meter for quantification (Supplementary Fig. S7).

Employing the toehold switch (N3 B) specific for the viral gene N (target region N3), we then demonstrated the highly sensitive and specific detection of SARS-CoV-2 (Fig. 5). Using genomic RNA isolated from the virus, coupled with an upstream NASBA amplication step, we found that the N3 sensor could detect as few as 100 RNA copies (1 μL of RNA at 166 aM, Fig. 5a). When this sensor was challenged with off-target viral RNA (1.66 pM) isolated from other respiratory viruses (influenza A subtypes H1N1 and H7N9, MERS), sensor activation and glucose production was highly selective for SARS-CoV-2 (Fig. 5b). Similarly sensitive and selective detection was also observed for the other sensor evaluated (gene E, Supplementary Fig. S8). As a final demonstration, we tested the N3 sensor using clinical RNA samples from six patients, in parallel with RT-qPCR (Fig. 5c). The glucose meter-based diagnostic results matched RT-qPCR performance, with clear discrimination of three SARS-CoV-2 positive patient samples (samples 4-6) from negative control patient samples (samples 1–3). It is worth highlighting that the assembly and prototyping of these sensors was done in a matter

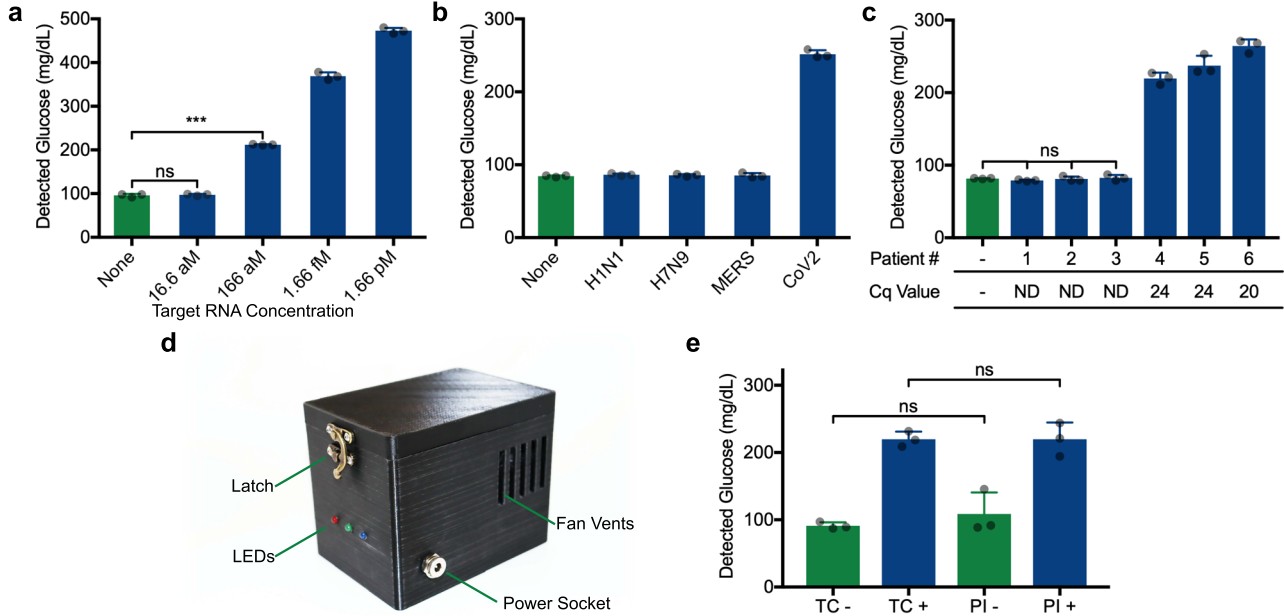

**Fig. 5 Demonstration of the SARS-CoV-2 gene N sensor coupled to glucose meter detection. a** Assay sensitivity (using N3 B toehold switch) demonstrated with purified viral RNA. The concentration indicated reflects the viral RNA concentration of the 1 μL aliquot added to NASBA reactions prior to glucose generation. ***$p = 0.0001$. **b** The SARS-CoV-2 gene N sensor (N3 B) specificity was tested using viral RNA genomes isolated from influenza A subtypes H1N1 and H7N9, MERS and SARS-CoV-2 using an initial RNA concentration of 1.66 pM. **c** Sensor N3 B performance upon exposure to RNA extracted from 6 different patient samples (3 SARS-CoV-2 positive and 3 negative patients) assayed with glucose-based sensor. Switch DNA template present at 1.25 ng/μL. Neg: no RNA sample added to NASBA. Cq values from the parallel RT-qPCR assay of these sample listed on the x-axis. ND: not detected. **d** Companion portable incubator. This device is capable of facilitating the experiment in place of a thermocycler. **e** Using sensor N3 B, direct comparison between the thermocycler (TC) and portable incubator (PI). SARS-CoV-2 RNA was added to positive (+) reactions at 1.66 pM. For **c**, statistical analysis is one-way ANOVA. **a** and **b** are comprised of 3 technical replicates, representative of 3 independent experiments. **e** is comprised of 3 biological replicates. ns: not significantly different. Data are presented as mean values ± SD. Source data are provided as a Source Data file.

of days, illustrating the potential for synthetic biology-based solutions to respond to urgent public health needs.

To match the portability of the glucose meter, we developed a companion portable incubator to provide temperature control at the point of use. This compact instrument is comprised of a temperature-resistant polycarbonate, 3-D printed outer shell that contains a heated aluminum lid and block controlled by a simple four-way switch (Fig. 5d). To operate, users simply set the switch to the corresponding incubation condition for lysis (100 °C), NASBA (65 °C then 41 °C) or cell-free reaction (37 °C). Using a thermoelectric Peltier element and a PTC heating element, temperature transitions between these settings is rapid, taking only 1–3 min. LEDs on the front of the device indicate heating (red), cooling (blue) and correct target temperature (green). Here we demonstrate the portable incubator (PI) in a side by side comparison with a lab-based thermocycler (TC) in the detection of SARS-CoV-2 viral RNA (Fig. 5e).

## Discussion

We have developed a robust and versatile system to interface gene circuit-based sensors with off-the-shelf glucose meters to create a new potential class of point-of-care diagnostic tools. The utility of this platform is confirmed through a series of proof-of-concept experiments that demonstrate accurate and sensitive detection of global health threats such as *S. typhi* and the SARS-CoV-2 virus in as little as two hours (Figs. 3–5). To tackle the challenge of endogenous glucose in patient samples, we presented three possible use case modes where such possible interference could be avoided with the use of programmable glucose reduction (Fig. 1e), sample dilution (Fig. 4) and nucleic acid purification (Fig. 5).

Taken together, we show that the glucose meter interface has the potential to distribute gene circuit-based sensing capacity outside of the laboratory. This capability could have near-term benefits for global health, especially in the context of the SARS-CoV-2 outbreak and the associated diagnosis bottlenecks across the globe. Much of the current gene circuit-based sensing capacity developed thus far is limited to lab-based assays. By developing this glucose meter interface, we hope to provide an accessible and low-cost means of distributing the capacity of cell-free gene circuit-based sensors to applications beyond the laboratory environment.

Our estimated cost per test is $9.26 USD for the commercially-available molecular components that comprise the assay and $0.40 USD for the glucose meter test strip. Molecular costs could be reduced significantly using in-house PURE system protocols that are estimated to be ~6-fold less expensive[51]. Perhaps most importantly, the mass production of glucose meters has brought the economy of scale to the device, which can be found for as little as ~$11 USD. This compares favorably to PCR instruments for molecular diagnostics, which are considerably more expensive and generally not portable. Challenges certainly remain, such as the universal diagnostic requirement for sample preparation; however we see the introduction of the glucose meter interface, and companion incubator, as a step toward a more decentralized model for diagnostics. Heat-based lysis and dilution, which we demonstrate for the detection of *S. typhi* (Fig. 4), has been demonstrated previously[4,52] and does seem to hold promise for sample preparation for low-resource applications.

In this context, and in combination with the established freeze-dried and biosafe CFS format for distribution[4,17,19], glucose meters hold the potential to distribute the detection of any analyte for which a gene circuit-based sensor can be designed. Through

pairing with gene circuit-based sensors, we see glucose meters as an already-in-place platform that could serve to monitor the presence of virtually any pathogen nucleic acid signature, genetic disease[4,5,17] and a wide range of small molecules (e.g., heavy metals[53], pesticides[54]) or other biomolecular targets[55,56]. Outside of health care, this de-centralized capacity has potential in education, agriculture, environmental monitoring and national security applications.

This work also accelerates an important trend in the field of synthetic biology, which is the integration of molecular technologies with hardware[38,57]. In this context, the cell-free gene circuit format is especially exciting as it allows engineered molecular components to directly interact with electronics, enabling the potential for purpose-built hybrid tools. As we have shown here, the coupling of hardware with molecular technologies has the potential to augment the utility of engineered molecular sensors, transforming them into broadly accessible tools. Moreover, the readily programmable nature of gene circuit-based sensors means that common microelectronic interfaces, like the glucose meter, could be used as universal readers without the need for tailor-made hardware infrastructure. Looking forward, we see the glucose meter as a potential vehicle that could serve as a gene-circuit interface for the field of synthetic biology as it transitions to increasingly practical applications.

## Methods

**Molecular assembly of gene circuits**. Reporter genes were assembled with upstream toehold switches or the TetO operator using overlap extension PCR with 20 bp overhangs between fragments. Where needed, gene circuits were cloned into pET15b using Gibson assembly[58]. For rapid assembly and screening of COVID-19 sensors, toehold devices were computationally designed[37] and synthesized as single-stranded ultramer oligonucleotides with a T7 promoter sequence at the 5' end and a common, homologous linker region on the 3' end for coupling to the *lacZ* reporter gene. Gene circuits were screened using CFS colorimetric activity assays (CPRG assay for lactase[4], GDH-NAD assay (described below) for trehalase, or PNPP assay for phosphatase[59]). Best-performing toehold switches were then tested for their glucose output using the glucose meter interface as described below. All genes, toehold switch targets, toehold switch sequences and primers are listed in Supplementary datasets 1–5.

**PURExpress reactions**. NEB PURExpress was used as the sole cell-free protein expression system in all experiments. Cell-free reactions where prepared according to the NEB PURExpress protocol (40% solution A, 30% solution B, along with DNA and other supplements) in a final volume of 8 μL. Supplemental components include 0.5% v/v RNase inhibitor (NEB M0314S) and 0.0125% v/v Tween-20 (BioShop TWN510.500). DNA encoding the molecular components of each reaction was added in the final concentrations specified for each experiment, along with the corresponding substrate for each enzyme. DNA was supplied as either a linear PCR product or plasmid. Reactions were assembled as a master mix and then aliquoted in 200 μL PCR tubes where the components specific to each treatment were added. Reaction volumes were brought up to 8 μL with nuclease-free water (Invitrogen #10977015).

For glucose meter measurements, reactions were incubated in a pre-equilibrated thermocycler at 37 °C for 1 h unless otherwise specified.

For plate reader colorimetric assays, a 384-well black clear-bottom plate (Corning # 3544) was used. Filter papers (Whatman #Z241067) for paper-based reactions were blocked in 5% BSA overnight, then dried[17]. A 2 mm biopsy punch (Miltex, VWR CA-95039-098) was used to cut a 2 mm disc from the prepared filter paper, which was then placed into reaction wells of the 384-well plate. Aliquots (1.8 μL) of the corresponding cell-free reactions were then added to paper discs in triplicate. Reactions were surrounded by a row of water-containing wells and the plate was then sealed with a clear plastic film (Sarstedt 95.1994) to reduce evaporation before being placed into the plate reader (BioTek, Fisher Scientific BTNEO2M). Gen5 3.02 software was used to operate the plate reader and collect the data.

**Tetracycline sensing**. To ensure tight control of expression from the TetO-Tre construct in treatments where TetR was included, reactions were assembled using CFS containing the TetR repressor pre-expressed in an overnight reaction (16 hr at 37 °C) from linear TetR-His[39] (30 nM) at 2% v/v final reaction volume. In reactions that did not contain TetR, the equivalent CFS without TetR-His was added (2%). All reactions contained 10 nM linear TetO-Tre DNA in the presence or absence of tetracycline (5 μM).

**Nucleic acid sequence-based amplification (NASBA) reactions**. Isothermal amplification of targets was performed using a commercial NASBA kit (Life Sciences Advanced Technologies NWK-1). Reactions were performed in 5 μL volume. Per reaction, 1.67 μL of reaction buffer, 0.83 μL of nucleotide mix, 0.05 μL of RNase inhibitors and 500 nM of each primer were mixed along with 1 μL of sample (water containing target RNA or sample lysate). Reactions were assembled at room temperature, incubated at 65 °C for 2 min, then 41 °C for 10 min before adding the enzyme mix (1.25 μL). This was followed by incubation at 41 °C for 1 h. Reactions were then added to the PURExpress reaction as described[17] at a 1/7 dilution for detection of amplified target(s).

**Glucose measurements with glucose meter**. Reactions were removed from the thermocycler and placed on ice. Each tube was flicked gently to ensure homogeneity. Using a pipette, 0.7 μL was drawn from incubated reactions and measured with the glucose meter (Bayer Contour Blood Glucose Monitoring System - Model 9545 C). For measurement, this volume was extruded from the pipette tip and then touched to the end of the glucose meter test strip to ensure even uptake and prevent bubbles. Technical replicates were performed by repeated measurement of the reaction with the glucose meter, in order to evaluate the ability of the glucose meter to yield consistent experimental results.

**Glucose measurements with the glucose dehydrogenase absorbance assay**. Following overnight incubation (37 °C), cell-free reactions containing the gene circuit of interest were added at 5% (final assay volume) to the GDH/NAD-based glucose assay buffer. This buffer contained 2 mM NAD, 2.6 U/mL GDH (Sigma-Aldrich #19359), 60 mM potassium phosphate pH 7.6 (reaction conditions adapted from Strecker[60]). For each of the reactions, a total reaction volume of 23.1 μL was prepared and 7 μL aliquots distributed in triplicate to a 384-well plate, without the use of paper disks. Wells were then monitored for absorbance at 340 nm for 1 h, with the final absorbance values used to compare treatments.

**Screen for typhoid-related toehold switch-based sensors**. CFS reactions containing each of the corresponding toehold switches were assembled as described above and then supplemented with 5 mM trehalose, 2.6 U/mL Glucose Dehydrogenase (Sigma-Aldrich #19359) and 10 mM β-NAD (BioShop NAD001). Linear DNA encoding toehold switches (1.25 ng/μL) and linear DNA (0.337 ng/μL) encoding the corresponding trigger RNA was added to the appropriate reactions. Reactions were then distributed in triplicate into a 384-well plate, as described above, and read for absorbance at 340 nm for at least 3 h in a plate reader.

**Screen for SAR-CoV-2-related toehold switch-based sensors**. The PURExpress cell-free reaction described above was supplemented with 0.6 g/L CPRG (chlorophenol red-b-D-galactopyranoside, Roche #10884308001) as the colorimetric substrate[4,17]. To demonstrate the potential for rapid sensor validation, unpurified linear PCR products encoding toehold switches (0.4 μL) and their corresponding trigger (0.4 μL) were added to 5 μL cell-free reactions. Single reactions were then distributed into a 384-well plate and monitored for absorbance at 570 nm in a plate reader (≥ 3 h).

**Online result interpreter**. User inputs the measured glucose concentrations of both a blank and sample. If the blank value is within the 95% confidence interval (CI) for a negative result (118 ± 15 mg/dL), the sample result is compared to the average negative value (118 mg/dL). The difference is used to evaluate the test result based on the 95% CI values for each possible result, based on experimental data. The other possible results are typhoid positive (149 ± 8 mg/dL), fluoroquinolone resistance positive (183 ± 9 mg/dL), and positive for both (237 ± 27 mg/dL). Code is available online (DOI: 10.5281/zenodo.4277743).

**Bacterial (*S. typhi*) cell lysis for RNA detection**. *S. typhi* cells (strain Ty2) were inoculated in LB broth from plated colonies and incubated at 37 °C overnight (approx. 16 h) to create a starter culture. This culture was then diluted (approx. 1/50) in fresh LB broth and incubated for approximately 2 h, after which OD600 measurements were taken. These measurements were used to calculate the approximate CFU/mL concentration of the culture using a conversion factor of $5 \times 10^8$ CFU/mL = 1 OD600. Cultures were then diluted to $10^6$ CFU/mL in 1 mL TE buffer (10 mM Tris-HCl, 1 mM EDTA, pH 7.5), and diluted again to $10^3$ CFU/mL with 10 mL 7% FBS in water. Negative controls were comprised of 10 mL 7% FBS and received an equivalent volume of TE buffer. These samples were then vortexed to ensure mixing and centrifuged at 14,000 xg for 5 min to pellet cells. The supernatant was removed in a single smooth motion to avoid loss of *S. typhi* cells. Subsequently, 1 mL of TE buffer was added to the pellet and mixed to resuspend the cells, then transferred to a 1.5 mL tube and centrifuged again for 1 min. The supernatant was carefully removed with a pipette and the pellet was recovered in 10 μL of TE buffer containing 0.2% Triton X-100. This volume was then transferred to a PCR tube and heated to 100 °C for 10 min in a thermocycler. 1 μL of this lysate was then added to the corresponding NASBA reactions containing STY primers.

**SARS-CoV-2 RNA extraction and qPCR.** Nasopharyngeal swab samples from clinically diagnosed COVID-19 negative and positive patients were obtained from the clinical diagnostics lab of MSH/UHN with approval from the Research Ethics Boards (REB #20-0078-E) of Mount Sinai Hospital in Toronto, Canada. RNA was isolated from 100 µL of sample using the Total RNA Purification Kit (Norgen Biotek) according to the manufacturer's protocol. RT-qPCR was performed using the 2019-nCoV: Real-Time Fluorescent RT-PCR kit (BGI) with 1 µL of extracted RNA in a 10 µL reaction analyzed on a BioRad CFX384 real-time PCR detection system.

**Portable incubator.** All the parts used in making the incubator were low-cost off-the-shelf components and electronics, except for the enclosure, which was 3D printed using polycarbonate filament due to its high heat resistance, and the aluminum block and lid, which were machined. The aluminum block has a 4×4 set of slots shaped to accommodate standard 0.2 mL PCR tubes. The incubator has four different temperature settings (37, 41, 65, and 100 degrees Celsius) which are selected by controlling the four respective switches, one of which may be turned on at a time. Once the desired temperature is set, the aluminum block is heated or cooled to that temperature using the Peltier element. A low-cost single trip point temperature sensor (TC622VAT) is used to enable controlled temperature control of the aluminum block. The temperature that is set by each switch is adjusted by changing the resistance on adjustable resistors. During the heating cycle the red LED turns on to indicate heating, while during the cooling cycle the blue LED turns to indicate cooling. When the set temperature is reached, the green LED turns on to indicate that the desired temperature has been reached. See Supplementary Figs. 9–13 for further details. Total cost of incubator is estimated at $120 CAD (approximately $92 USD), excluding power supply and labor costs (see supplementary BoM).

**Statistical analysis.** Unless otherwise indicated, experimental data sets were compared using two-tailed Welch's $t$-test. All statistical analysis and graphing was done using Graphpad Prism 7 software.

## Materials
**General materials.** Unless otherwise noted, all chemicals were purchased from Sigma-Aldrich (St. Louis, MO, USA). DNA oligonucleotides were purchased from Integrated DNA Technologies (Coralville, IA, USA) and used without further purification. Cell-free expression buffers and commercial restriction enzymes were purchased from New England Biolabs (NEB, Ipswich, MA, USA). NASBA kits for isothermal amplification were purchased from Life Sciences Advanced Technologies Inc. (St Petersburg, FL, USA). Enzyme substrates were purchased from BioShop Canada Inc. (Burlington, ON, Canada): lactose (LAC234), trehalose (TRE222), glucose 6-phosphate (GPS095) NAD (NAD001).

**Enzyme genes.** The gene for lactase was sourced from Pardee et al. (2014)[17]. The gene for the glucose phosphatase (*E. coli*, *ybiV*) was provided by the Yakunin lab at University of Toronto. Trehalase gene (*C. japonicus*, *tre37a*) was acquired from DNASU (Arizona State University), contributed by Kelley Moremen (University of Georgia, NIH grant RR005351). Glucose dehydrogenase mutant gene (*B. subtilis*, variant E170K/Q252L)[36] was synthesized by IDT.

**Reporting summary.** Further information on research design is available in the Nature Research Reporting Summary linked to this article.

## Data availability
All data generated or analyzed during this study are included in a supplementary Source Data file, and available from the corresponding author upon request. Source data are provided with this paper.

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

## Acknowledgements

We thank Shana Kelley for advice and guidance, Anna Khusnutdinova and Alexander Yakunin for providing phosphatases to screen, Kelly Moremen for generating the trehalase clone used (NIH grant RR005351) and the Canadian Food Inspection Agency for providing the DNA oligos for SARS-CoV-2 work. This work was supported by funds to K.P. from the NSERC Discovery Grants Program (RGPIN-2016-06352), CIHR Foundation Grant Program (201610FDN-375469), CIHR Canada Research Chair Program (950-231075), University of Toronto's Medicine by Design initiative which receives funding from the Canada First Research Excellence Fund (C1TPA-2016-06), Canada's International Development Research Centre (grant 109434-001) through the Canadian 2019 Novel Coronavirus (COVID-19) Rapid Research Funding Opportunity and the University of Toronto COVID-19 Action Initiative 2020. This work was also supported by funds to A.A.G. from an NIH Director's New Innovator Award (1DP2GM126892), an NIH U01 award (1U01AI148319-01), the Gates Foundation (OPP1160667), an Arizona Biomedical Research Commission New Investigator Award (ADHS16-162400), an Alfred P. Sloan Fellowship (FG-2017-9108), and Gordon and Betty Moore Foundation funds (#6984) and an NIH R21 award (1R21AI136571-01A1) to K.P./A.A.G. The web interface was developed by BioHues Digital. The views, opinions, and/or findings contained in this article are those of the authors and should not be interpreted as representing the official views or policies, either expressed or implied, of the NIH.

## Author contributions

E.A. developed concept, designed and performed molecular experiments and co-wrote the manuscript; M.K. designed and performed isothermal amplification and SARS-CoV-2 experiments and co-wrote the manuscript, M.N. designed and performed SARS-CoV-2 experiments and co-wrote the manuscript. K.J. designed and performed endpoint colorimetric experiments and TetR experiments; P.S.M. performed antibiotic resistance gene sensing experiments other than the fluoroquinolone resistance ones. L.T. assisted in early reporter enzyme screening experiments. S.C and F.M. designed and built the portable heater device, with F.M. creating figures S9-S13. Y.G. provided code for the associated web interface. A.S. and J.H.B. contributed S. typhi cultures and related expertise, D.M. and A.A.G. designed the toehold switches. J.D.P., D.T., J.W., R.B. and T.M. provided the SARS-CoV-2 patient samples used, with J.D.P. performing the corresponding RT-qPCR analysis. M.P., A.A., S.B. and B.P. provided the purified SARS-CoV-2, influenza and MERS viral RNAs used. D.S., J.H.B. and A.A.G. supervised elements of the project. K.P. was responsible for project design and supervision, designed experiments and co-wrote the manuscript. All authors discussed the results and commented on the manuscript.

## Competing interests

The Governing Council of the University of Toronto has filed a patent application covering the invention of the gene circuit-glucose meter interface with inventors Evan Amalfitano, Margot Karlikow and Keith Pardee (International application number PCT/CA2018/051646, provisional status). Arizona State University has filed a patent application covering the invention of the SARS-CoV-2 sensors with Masoud Norouzi, Margot Karlikow, Alexander Green and Keith Pardee (US patent number 63/038,609, provisional status). The remaining authors declare no competing interests.
