## [Peer Review File · Nature Communications]

Reviewers' Comments:

Reviewer #1:

Remarks to the Author:

This manuscript by Amalfitano, Karlikow, Norouzi, and colleagues demonstrates a biological circuit that can convert a genetic signal to a glucose output, which can be read by a glucose meter. The authors demonstrate detection of small molecules, as well as diagnostics for typhoid, paratyphoid, and SARS-CoV-2.

This is a clever idea, and I like the paper overall. There is a good amount of novelty, and I like that the authors demonstrate their method works on a variety of assays, with solid performance across the board. However, I think the manuscript could benefit from some revisions before publication, mostly in terms of data presentation and framing. With improved presentation, I think this will be a very strong paper!

Major points:

1. The use of bar plots with a y axis that does not start at zero should be avoided. It is very misleading to use bar plots that do not start at $y=0$, as the height of the bars is no longer proportional to the values on the axes. The authors need to change the way there are presenting their data. If the authors want to continue using y axes that don't start at zero, dot plots (or something similar) should be used instead of bars.
2. The approach of using glucose as an output is very clever. It would be great to see some additional discussion of the cost (per machine, and per test), and the speed, of a glucose readout compared to other readouts. The authors allude to some of these points in the discussion, but more concrete numbers would really help make the case for the authors' approach.

Minor points:

3. In Fig 4., the language surrounding these results is confusing. These appear to be bacterial cultures diluted into serum - the use of the word "sample" implies that they may have come from a patient, which appears to not be the case.
4. Also related to Fig 4: is 1,000 CFU/mL clinically relevant? The reference (45) cited by the authors uses units of copies per mL, and not CFU per mL.

Additional comments:

The paper is well-written, and really strong conceptually.

Reviewer #2:

Remarks to the Author:

Review of "Addressing the Diagnostic Bottleneck: A Glucose Meter Interface for Point-of-Care SARS-CoV-2 and *S. typhi* Detection" For Nature Communications, by Amalfitano, et al.

The authors report the development of a nucleic acid assay that uses an off the shelf glucose meter for detection. They demonstrate this for detection of the product of a gene circuit sensor that senses RNA sequences for typhoid, paratyphoid, *Salmonella typhi*, and SARS-CoV-2. The assays uses nucleic acid amplification and a cell free system (CFS) for gene expression of an enzyme that can convert glucose-containing precursors into glucose monomers. This is achieved by a genetic circuit that uses a toehold switch for a trehalose reporter. These glucose monomers are then detected by the glucometer.

This approach could be used to detect RNA for different diseases. However, the novelty of the work is not completely clear, and there are several shortcomings that need to be addressed before publication. Furthermore, the manuscript is disorganized and needs to be improved significantly for readability.

1.) First, the novelty of the work needs to be articulated more clearly in the context of what has been reported before in the literature. The use of glucose meters to detect nucleic acid products has been reported before. Du et al. use a glucose meter to detect MERS (Scientific Reports, 2015 (DOI: 10.1038/srep11039), Zhang et al. use a glucose meter to detect the outputs of logic gates (Ange. Chemie, 2018, DOI: 10.1002/anie.201804292). The Du and Zhang papers ought to be cited. Because glucometer detection of nucleic acid products for diagnostics have been reported before, what is the main innovation of the work here? This needs to be more clearly stated. Otherwise, it seems like it is an extension of other work in synthetic biology, and lacks a broad impact.

2.) The paper is incoherent and meandering in its reporting of the results, and is really hard to follow. The title says the system they are investigating is for "S. Typhi and SARS-CoV-2" detection but there are a lot of experiments included (AmpR, TC, etc.) which are not tied directly to the main results, and thus are distracting. In addition, the reported results do not build upon another. For example, the authors vary the substrate concentration to tune the glucose output, but not for the ultimate system of interest (the S. Typhi and SARS-CoV-2 detection).

Also, the sensor in Figures 2e,f demonstrates functionality of an unrelated system, i.e., tetracycline detection, which doesn't really contribute to the main results in that it is a small molecule and not RNA, and the ultimate sensor is not reliant upon this switch functioning.

Furthermore, the SARS-CoV-2 results seem like they are sort of slapped in there at the last minute, as they are not the result of a logical path that is concretely built upon the experiments in the previous figures.

Thus, the authors need to integrate their results coherently and use them to support their claims, otherwise the manuscript seems like an assembly of random results.

To demonstrate novelty and applicability of their approach as a platform, and to have the results broadly appreciated, the authors should choose a single diagnostic target and coherently address the issues associated with detecting that target. Demonstration of simply the use of the glucose meter to detect the signal is not novel enough in light of already published reports in the literature. Even though technically one can swap out the sequence in the switch to detect different sequences, the associated issues such as what is the sample matrix, how the sample is collected, patient issues, etc. will differ depending on the disease of interest. For example, the mode of obtaining samples and the sample matrix for Salmonella are incredibly different from SARS-CoV-2—one is a bacteria in food, and the other a virus in people in swabs.

3.) Figure 1a includes a schematic that refers to colorimetric sensing on paper sensing, which is an extension of the authors Cell 2016 paper where they use this to detect zika sequences (Ref. 17). Because paper based colorimetric sensing is not presented in this manuscript, the figure should be modified to remove this.

4.) One issue that is not sufficiently addressed is that different patients will have different basal levels of glucose, which will contribute to the signal read out by the glucose meter. This is going to be a critical factor if this approach is used to test patients. How much variability is there? how much does glucose from the patient sample contribute to the signal? These need to be quantified for practical use of this as a diagnostic.

5.) The title states that the approach addresses a diagnostic bottleneck, but what bottleneck does

it specifically address? While the approach does take advantage of the availability and convenience of glucose meters, the assay still fundamentally requires a nucleic acid amplification step (nucleic acid sequence-based amplification, NASBA) and also expression by a cell free system (CFS). Both of these steps are not trivial to achieve, as they require lab instrumentation with temperature control and some sort of liquid handling, have strict purity issues, and cannot be done point of care. Figure 4 greatly oversimplifies the process—CFS is not even included in the schematic! The assay is not simply reading something out with a glucose meter, and the glucose meter is introduced at the final step. Furthermore, CFS kits are quite expensive. This is especially so since the assay requires a recombinant system (NEB PurExpress), and cannot use cell lysate kits because of background. These factors severely limit the deployability of the assay. The title states that the approach addresses a diagnostic bottleneck, but what bottleneck does it specifically address? It would be helpful if they include a description of the cost and the infrastructure needed to run an assay, with a schematic of the workflow, as the required instrumentation and technical expertise is not trivial.

7.) For a diagnostic approach, the assay needs to show proper validation, where it is compared against a gold standard with proper statistics, and true positive/negative rates are reported.

8.) What is required to go from sample to answer? In particular, what are the sample extraction steps that must be done? The authors need to include description of how realistic sample matrices would be handled.

9.) The description of how the AmpR and SptR switches is vague, and needs more detail.

10.) The enzymes in Figure 1b should be labeled with Tre, Lac, Phos, to help the reader understand what the enzymes are.

Dear Dr Pardee,

Your manuscript entitled "Addressing the Diagnostic Bottleneck: A Glucose Meter Interface for Point-of-Care SARS-CoV-2 and *S. typhi* Detection" has now been seen by 2 referees, whose comments are appended below. You will see from their comments copied below that while they find your work of considerable potential interest, they have raised quite substantial concerns that must be addressed. In light of these comments, we cannot accept the manuscript for publication, but would be interested in considering a revised version that addresses these serious concerns with a focus on SARS-CoV-2 diagnosis.

We hope you will find the referees' comments useful as you decide how to proceed. Should further experimental data or analysis allow you to address these criticisms, we would be happy to look at a substantially revised manuscript. However, please bear in mind that we will be reluctant to approach the referees again in the absence of major revisions.

We would normally ask to see a revised version of this paper within 3 months but we appreciate revisions may take longer than usual and can extend this timeline if the Covid-19 pandemic prevents you from undertaking any further work for a longer period - please do get back to us on this nearer the time.

We are committed to providing a fair and constructive peer-review process. Do not hesitate to contact us if you wish to discuss the revision or if there are specific requests from the reviewers that you believe are technically impossible or unlikely to yield a meaningful outcome.

When resubmitting, you must provide a point-by-point response to the reviewers' comments. Please show all changes in the manuscript text file with track changes or colour highlighting. If you are unable to address specific reviewer requests or find any points invalid, please explain why in the point-by-point response.

In addition to the above, you must comply with the following editorial requests; failure to do so will cause delays upon resubmission.

Reply to Reviewers: NCOMMS-20-12233A-Z

"A Glucose Meter Interface for Point-of-Care Gene Circuit-based Diagnostics"

We thank the Reviewers for their enthusiastic, constructive and thoughtful comments, which have led to a significantly improved manuscript. Below, we address all of the Reviewers' comments in detail and describe the experiments that have been performed during the revision process and included in the revised manuscript.

REVIEWER COMMENTS

Reviewer #1

Comment:

This manuscript by Amalfitano, Karlikow, Norouzi, and colleagues demonstrates a biological circuit that can convert a genetic signal to a glucose output, which can be read by a glucose meter. The authors demonstrate detection of small molecules, as well as diagnostics for typhoid, paratyphoid, and SARS-CoV-2.

This is a clever idea, and I like the paper overall. There is a good amount of novelty, and I like that the authors demonstrate their method works on a variety of assays, with solid performance across the board. However, I think the manuscript could benefit from some revisions before publication, mostly in terms of data presentation and framing. With improved presentation, I think this will be a very strong paper!

Response:

We thank the Reviewer for their comments and for highlighting the strength of the paper, including our efforts to demonstrate the technology across a range of assays. We are also grateful to the Reviewer for noting the need to improve the presentation of some data. As we will discuss below, we have now made these changes.

Comment:

Major points:

1. The use of bar plots with a y axis that does not start at zero should be avoided. It is very misleading to use bar plots that do not start at $y=0$, as the height of the bars is no longer proportional to the values on the axes. The authors need to change the way there are presenting their data. If the authors want to continue using y axes that don't start at zero, dot plots (or something similar) should be used instead of bars.

Response:

Thank you for highlighting this point and we agree that the graphs should have originally been designed to have the Y-axis begin at a value of zero. To address this, we have placed the Y-axis origin for the graphs of all glucose meter data set figures at zero. This includes modification of Fig. 1 d, Fig. 2, Figs. 3a-d, Fig. 4b, Fig. 5 and Fig. S7.

Minor notes: Since a couple experiments during the technology development stage had higher signal background as a result of the short 1-hour time point*, we have moved Figs. 2 c,d to the supplementary section (Figs. S1 a,b) . In another case (Fig. S2), we have left the Y-axis with an origin value of 2.0; however, this data includes a time series of measurements which allows for a comparison between all data sets. We would like to ensure that the data is presented as transparently as possible, and so we are happy make any additional adjustments to figures if the Reviewer feels there is a need. Thank you.

*To provide some context, the background signal that is observed in some experiments is not the result of glucose in the cell-free reaction, but rather a non-specific effect from short time course experiments. This background is consistent between batches and is likely the result of a component in the PURExpress reaction that interacts with the glucose meter electrode, generating an electrochemical signal. This background completely disappears after an overnight incubation, as can be seen in Fig. 1c.

Comment:

2. The approach of using glucose as an output is very clever. It would be great to see some additional discussion of the cost (per machine, and per test), and the speed, of a glucose readout compared to other readouts. The authors allude to some of these points in the discussion, but more concrete numbers would really help make the case for the authors' approach.

Response:

We appreciate the reviewer highlighting these points as these are features that may be of interest to the readers. In addition to the portability afforded by the glucose meter-based interface, the diagnostic cost and speed of the glucose meter interface are important advantages that we had not highlighted in

the original draft of our manuscript. This information has now been added and can be found in the discussion on page 11.

Briefly, the cost of gene circuit-based sensor reaction using the indicated commercially-available molecular reagents is estimated to be \$9.26 (USD), and this could in principle be reduced to \$3.65 (USD) if in-house PURE-based reaction used (\$0.09/uL)¹.

As highlighted by the Reviewer, the instrumentation cost between these methods also could provide a significant advantage. The cost of the glucose meter used in this study (Contour, Bayer) starts at ~\$11 USD and is portable. This compares favorably to the RT-qPCR instrument used as the gold standard molecular diagnostic for SARS-CoV-2 patient sample testing (Fig. 5), which costs significantly more and requires a laboratory setting.

Minor points:

Comment:

3. In Fig 4., the language surrounding these results is confusing. These appear to be bacterial cultures diluted into serum - the use of the word "sample" implies that they may have come from a patient, which appears to not be the case.

Response:

Thank you. We agree that there was potential for misunderstanding in the Figure legend text as originally written. We have now added the text "from serum-containing mock samples" to replace "from serum-containing samples", and we have changed the second instance of "sample" to now read "from the serum-containing input containing 1000 CFU/mL" (Figure 4 legend, page 9).

Similarly, in the main text (page 9), in one case, we have removed the word "sample" so that the sentence now reads "... detection of endogenous RNA from whole-cell *S. typhi*". In another case, we added the word "mock" so that the sentence now reads "from serum-containing mock samples with 10³ CFU/mL *S. typhi*"

Comment:

4. Also related to Fig 4: is 1,000 CFU/mL clinically relevant? The reference (45) cited by the authors uses units of copies per mL, and not CFU per mL.

Response:

Thank you and we apologize for the confusion. We cited Massi *et al.* (2005)² who used TaqMan-based real-time PCR to quantify the *S. typhi* flagellin gene in blood culture-positive patient samples. Their findings report that "blood samples with a positive blood culture had *S. typhi* loads ranging from 1.01×10³ to 4.35×10⁴ DNA copies/mL".

Here we demonstrate that the glucose meter interface can detect *S. typhi* at a concentration of 1000 colony forming units (CFU)/mL. While the conversion of CFU/mL to DNA copies/mL may not be 1:1 in patient samples (e.g. potentially dead cells in patient circulation), here we use cultured *S. typhi* collected in the exponential phase of growth and, as such, the cells are likely to have a high level of viability. With this in mind, we feel that detection by the glucose meter from samples containing 1000 CFU/mL places the sensitivity of the system within the clinically relevant range of blood culture-positive samples (1000 – 43,500 DNA copies), as defined by Massi *et al.*

We see the glucose meter-based detection of *S. typhi* as providing significant advantages over the lab-based diagnostic methods of blood culture and PCR in low-resource setting. We have modified the text to clarify this point (page 9).

Comment:

Additional comments:

The paper is well-written, and really strong conceptually.

Response:

We thank the Reviewer for their positive feedback.

Reviewer #2 (Remarks to the Author):

Review of “Addressing the Diagnostic Bottleneck: A Glucose Meter Interface for Point-of-Care SARS-CoV-2 and *S. typhi* Detection” For Nature Communications, by Amalfitano, et al.

Comment:

The authors report the development of a nucleic acid assay that uses an off the shelf glucose meter for detection. They demonstrate this for detection of the product of a gene circuit sensor that senses RNA sequences for typhoid, paratyphoid, Salmonella typhi, and SARS-CoV-2. The assays uses nucleic acid amplification and a cell free system (CFS) for gene expression of an enzyme that can convert glucose-containing precursors into glucose monomers. This is achieved by a genetic circuit that uses a toehold switch for a trehalose reporter. These glucose monomers are then detected by the glucometer.

This approach could be used to detect RNA for different diseases. However, the novelty of the work is not completely clear, and there are several shortcomings that need to be addressed before publication. Furthermore, the manuscript is disorganized and needs to be improved significantly for readability.

1.) First, the novelty of the work needs to be articulated more clearly in the context of what has been reported before in the literature. The use of glucose meters to detect nucleic acid products has been reported before. Du et al. use a glucose meter to detect MERS (Scientific Reports, 2015 (DOI: 10.1038/srep11039), Zhang et al. use a glucose meter to detect the outputs of logic gates (Angewandte Chemie, 2018, DOI: 10.1002/anie.201804292). The Du and Zhang papers ought to be cited. Because glucometer detection of nucleic acid products for diagnostics have been reported before, what is the main innovation of the work here? This needs to be more clearly stated. Otherwise, it seems like it is an extension of other work in synthetic biology, and lacks a broad impact.

Response:

We thank the Reviewer for mentioning these additional publications and we apologize for the oversight. We have now added these citations, along with further text to the manuscript to clarify the innovative nature of our approach (please see page 2).

Briefly, as the Reviewer highlights, previous work to re-purpose personal glucose meters for the detection of analytes other than glucose has been reported. Importantly, the work reported by the Du and Zhang papers, along with other publications from the Lu lab, rely on a molecular mechanism that is fundamentally different from the system that we report here. In these reports, the diagnostic is distributed to end users with a DNA-invertase conjugate bound to magnetic beads and, if the analyte

of interest present, the enzyme is released from the bead, and following a bead removal and substrate addition step, the glucose is produced in response.

In contrast, in the method we present here, the glucogenic reporter enzyme is synthesized *de novo* in response to a target analyte. This approach is fundamentally different at the molecular level and conveys several advantages.

- i. **Universal output platform for the field of synthetic biology.** In principle, the glucose meter interface we present can be used by any gene circuit-based sensor from the field of synthetic biology that creates a reporter protein output. This makes the approach broadly useful to the field in general, and with the emerging importance of gene circuit-based sensors in diagnostics, it provides an important new way for the field to place diagnostic capacity into the hands of users with an off-the-shelf device that already has clinical acceptance.
- ii. **Low-cost development.** The gene circuit-based interface that we present can be assembled directly from unmodified DNA, which is inexpensive and scalable. This is an important distinction for a diagnostic tool, and one that has the potential to enable low-cost and accessible diagnostics.

In contrast, while the DNA-invertase bead-based method is exciting, implementation of the approach requires expensive chemically-modified DNA inputs and the creation of relatively complex DNA-protein hybrid molecules that are conjugated to beads. Such engineered materials can be difficult and costly to manufacture, which may limit testing and implementation of the technology in practice, particularly in the resource-limited settings where we envision these tests being deployed.

Moreover, the DNA-invertase conjugates likely require a constant cold chain for stable storage and distribution, which can be costly. We have previously demonstrated that the underlying cell-free diagnostic platform that we present here can be freeze-dried for room temperature storage and distribution^{3,4}.

- iii. **Simple use case workflow and handling.** In the method that we report, the glucogenic reporter enzyme is synthesized on-site in the presence of the correct target analyte. This provides a simple workflow that minimizes user error and the risk of false positive results under practical conditions.

In contrast, the invertase bead-based method requires complex workflow where users must carefully remove all invertase containing beads to avoid false positive results.

Comment:

2.) The paper is incoherent and meandering in its reporting of the results, and is really hard to follow. The title says the system they are investigating is for “S. Typhi and SARS-CoV-2” detection but there are a lot of experiments included (AmpR, TC, etc.) which are not tied directly to the main results, and thus are distracting. In addition, the reported results do not build upon another. For example, the authors vary the substrate concentration to tune the glucose output, but not for the ultimate system of interest (the S. Typhi and SARS-CoV-2 detection).

Response:

We welcome the Reviewer’s critique and the opportunity to improve the readability of the manuscript. In response, we have reviewed the manuscript for text where we can clarify the message with additional text and subheadings.

We began with a change to the title of the manuscript to better capture that this manuscript presents the development of a technology that can be applied to a diverse array of applications. The title now reads “A Glucose Meter Interface for Point-of-Care Gene Circuit-based Diagnostics”. We hope that the change of title clarifies this point for the readers.

The narrative of the manuscript strives to first describe the technical innovations required to establish the technology and then goes on to demonstrate the detection of a series of diverse and challenging analytes to illustrate the versatility of the approach and its potential impact.

Here we outline the progression of the manuscript, with reference to the related data and where we have modified the text to improve clarity.

- i. **Technology development.** We began by first describing the innovations required to develop the glucose meter interface for gene circuits. This includes the discovery of robust glucogenic reporter enzymes (Fig. 1c,d), programmed glucose reduction (Fig. 1e), the optimization of buffer conditions, as well as substrate and DNA concentrations (Fig. 2a,b).
- ii. **Demonstration of RNA sensing** (Fig. S1).
- iii. **Demonstration of small molecule detection** (Fig. 2c,d).
- iv. **Demonstration of the potential for the glucose meter interface in diagnostic applications** with the detection of *S. typhi*, paratyphoid A, B and the related drug-resistance gene fluoroquinolone (Figs. 3 and 4).
- v. **Demonstration of the potential for rapid diagnostic development** with the detection of the SARS-CoV-2 RNA genome from RNA, cultured virus and patient samples (Fig. 5).

Importantly, while we have endeavored to address the Reviewer #2’s comment, we also received feedback from Reviewer #1 who found the paper to be “**well-written, and really strong conceptually**”. With this in mind, we have opted to respond to the comment above in a focused manner rather than with sweeping changes.

Comment:

Also, the sensor in Figures 2e,f demonstrates functionality of an unrelated system, i.e., tetracycline detection, which doesn’t really contribute to the main results in that it is a small molecule and not RNA, and the ultimate sensor is not reliant upon this switch functioning.

Response:

Thank you for highlighting that the inclusion of tetracycline detection did not seem to contribute to demonstrations with *S. Typhi* and SARS-CoV-2 detection. We hope that the change of title and our changes to the text (pg. 5) now clarifies that this data is meant to demonstrate that the potential range of application of the glucose meter interface includes small molecule detection (e.g. tetracycline).

Comment:

Furthermore, the SARS-CoV-2 results seem like they are sort of slapped in there at the last minute, as they are not the result of a logical path that is concretely built upon the experiments in the previous figures.

Thus, the authors need to integrate their results coherently and use them to support their claims, otherwise the manuscript seems like an assembly of random results.

To demonstrate novelty and applicability of their approach as a platform, and to have the results broadly appreciated, the authors should choose a single diagnostic target and coherently address the issues associated with detecting that target. Demonstration of simply the use of the glucose meter to detect the signal is not novel enough in light of already published reports in the literature. Even though technically one can swap out the sequence in the switch to detect different sequences, the

associated issues such as what is the sample matrix, how the sample is collected, patient issues, etc. will differ depending on the disease of interest. For example, the mode of obtaining samples and the sample matrix for Salmonella are incredibly different from SARS-CoV-2—one is a bacteria in food, and the other a virus in people in swabs.

Response:

Changes to the title, sub-titles and text of the manuscript described above will hopefully clarify for the reader that the work presented has been assembled to describe and demonstrate a new platform technology for the field of synthetic biology. Further, we expect that these changes to the manuscript will make clear that the presented applications of the glucose meter interface are intentionally broad to highlight the potential range of use.

In response to Reviewer's suggestion of demonstrating the technology with only one diagnostic target, we feel that the demonstration of typhoid detection is important to highlight the potential of the glucose meter interface to bring affordable and accessible molecular diagnostics to low- and middle-income countries. In these settings, the lab-based nature of conventional blood culture- and PCR-based methods make the typhoid detection a longstanding challenge. We also feel the current global COVID-19 public health crisis, where access to diagnostics has been a front-page issue since it began, makes the demonstration of SARS-CoV-2 detection incredibly topical and of broad interest to readers. Taken together, these two diagnostic targets also demonstrate the detection of both bacterial and viral pathogens, further showing range of use.

In response to concerns over novelty, the platform we present is very distinct from the molecular mechanism in previous work reported by Du, Zhang and other Lu lab efforts, and this distinction has important practical implications (as described above). Further, the programmable nature of gene circuit-based sensors in general means that, by simply changing the DNA sequence, the glucose meter interface we describe has the potential to enable the detection of small molecules (e.g. heavy metal, contaminants), bacterial diseases, viral pathogens or any other gene of interest. As we describe above, the DNA-invertase-bead-based system reported in these previous publications requires the development of custom DNA-protein conjugates to respond to each upstream aptamer output sequence.

The Reviewer correctly highlights that the sample preparation for different sensing applications may differ; however, this is a universal challenge faced by all diagnostic modalities including RT-qPCR, CRISPR-based diagnostics like SHERLOCK and DETECTR, and the cited work by Du and Zhang. The work we present is a proof-of-concept report of a new detection and diagnostic reading technology. There is not a precedent for new diagnostic technology development manuscripts also tackling sample preparation in the initial publication. Such efforts are generally developed at a later time, for example, as part of a patient trial.

There are also many other previously published studies that focus solely on solving the sample preparation challenge, such as HUDSON⁵, which was published as a subsequent report following the initial two SHERLOCK manuscripts. We agree with the Reviewer on the importance of such practical details for implementation and this is very much an active challenge for the field of point-of-care diagnostics. Our report of this glucose meter interface contributes to this challenge and brings the practical deployment of molecular diagnostics into the hands of users, one step closer.

Related to the challenge of implementing point-of-care diagnostics, we have introduced a new component to the manuscript that allows for the portable incubation of all diagnostics steps. This small device can be battery-operated and facilitates the pathogen lysis, isothermal amplification and toehold switch-based reaction steps (Fig. 5d). The compact device is comprised of an aluminum block heated under the control of simple four-way switch. The CAD files and circuit diagrams are included in the supplementary information for reader who may want to replicate the device (Figs S9-

S13). We demonstrate the portable incubator with detection of SARS-CoV-2 viral RNA in an experiment that compares the side by side performance of the incubator with a conventional thermocycler (Fig. 5e). This additional step toward the practical implementation of our technology is paired with the web-based interface that we originally included for the conversion of glucose measurement for the detection of *S. typhi*. Here we also introduce an updated and more user-friendly interface for users to convert glucose values (Fig. 3d).

Comment:

3.) *Figure 1a includes a schematic that refers to colorimetric sensing on paper sensing, which is an extension of the authors Cell 2016 paper where they use this to detect zika sequences (Ref. 17). Because paper based colorimetric sensing is not presented in this manuscript, the figure should be modified to remove this.*

Response:

Thank you for the feedback. We have removed the colorimetric feature from the schematic. It had originally been included as a point of comparison for readers between the glucose meter approach and conventional optical gene circuit outputs in the field of synthetic of biology.

Comment:

4.) *One issue that is not sufficiently addressed is that different patients will have different basal levels of glucose, which will contribute to the signal read out by the glucose meter. This is going to be a critical factor if this approach is used to test patients. How much variability is there? how much does glucose from the patient sample contribute to the signal? These need to be quantified for practical use of this as a diagnostic.*

Response:

We appreciate the Reviewer raising this point. We have given this feature of the glucose meter interface considerable thought. We have added text to the discussion on page 11 to address how this potential challenge in the implementation of the technology could be met. Below we outline how we see the glucose meter interface operating in the context of endogenous glucose in patient samples.

To begin, the concentration of glucose in healthy patients is 72-140 mg/dL and this range can extend from 50 mg/dL to 200 mg/dL in hypoglycemia and hyperglycemic patients, respectively. An answer to the Reviewer's question depends on the approach to sample preparation and below we describe three common scenarios.

- i. **Nucleic acid detection using purified RNA/DNA.** For RT-qPCR and other molecular diagnostic methods, the most common approach to sample preparation uses RNA/DNA isolated from the patient sample matrix. This is typically done using silica-based extraction. In this scenario, the endogenous glucose from patients is completely washed away and is not a factor in the use of the glucose meter interface. In the work we present, this potential approach was demonstrated in detection of SARS-CoV-2 from patient samples (Fig. 5c).
- ii. **Nucleic acid detection using sample dilution.** Since the isolation of nucleic acid from a sample is not always practical, another approach to sample preparation for molecular diagnostics can use dilution of the original sample to reduce the concentration of potential inhibitors below the critical threshold. This approach has proven to be broadly useful in a wide range of molecular diagnostic applications⁶⁻⁸.

In some the work we present, the glucose concentration of a sample would be diluted through the course of the experiment. For example, in the detection of target sequence directly from *S. typhi* (Fig. 4), the sample is first added to the NASBA reaction as 1/5 of the final volume (5x dilution). The NASBA reaction after amplification is then added to the CFS as 1/7 the final volume (7x dilution, for a total of 35x from the original sample). With an overall dilution of

35x, the contribution of glucose from healthy (106 mg/dL) and hyperglycemic (200 mg/dL) patients would be to be 3 mg/dL and 5.7 mg/dL, respectively. As can be seen from the Y-axes of our diagnostic demonstrations, even a contribution of 5.7 mg/dL from hyperglycemic patients would not influence the diagnostic outcome (Figs. 4,5).

We have also previously demonstrated that the toehold switch-based sensor type that we used here can detect pathogen (i.e. Zika virus) directly from viremic plasma samples collected from rhesus macaques using this dilution-based strategy⁴.

- iii. **Detection of small molecules from a patient.** There are potential applications where the sensing of metabolites or other small molecules from patients may not allow for detection from diluted samples. To address this potential challenge, here we report the development of a method for the programmable removal of background glucose from patient samples (this background can be easily determined using a conventional glucose meter measurement).

As we describe on pages 3 and 4, to remove a pre-set amount of glucose from a sample, we have exploited the enzyme glucose dehydrogenase (*B. subtilis*, variant E170K/Q252L⁹) that converts glucose into glucono-1,5 lactone, which is inert to the glucose meter. The enzyme requires one molecule of NAD for every molecule of glucose that is catabolized. As we show, with this information in hand, we were able to develop a method that could selectively clear fixed amounts of glucose from incoming samples (Fig 1e).

We feel that these three scenarios collectively represent a relatively comprehensive set of use cases for the technology and that the work we present demonstrates a practical path forward for the technology.

Comment:

5.) The title states that the approach addresses a diagnostic bottleneck, but what bottleneck does it specifically address? While the approach does take advantage of the availability and convenience of glucose meters, the assay still fundamentally requires a nucleic acid amplification step (nucleic acid sequence-based amplification, NASBA) and also expression by a cell free system (CFS). Both of these steps are not trivial to achieve, as they require lab instrumentation with temperature control and some sort of liquid handling, have strict purity issues, and cannot be done point of care. Figure 4 greatly oversimplifies the process—CFS is not even included in the schematic! The assay is not simply reading something out with a glucose meter, and the glucose meter is introduced at the final step. Furthermore, CFS kits are quite expensive. This is especially so since the assay requires a recombinant system (NEB PurExpress), and cannot use cell lysate kits because of background. These factors severely limit the deployability of the assay. The title states that the approach addresses a diagnostic bottleneck, but what bottleneck does it specifically address? It would be helpful if they include a description of the cost and the infrastructure needed to run an assay, with a schematic of the workflow, as the required instrumentation and technical expertise is not trivial.

Response:

Thank you for raising these points. We address each below and have added text, as indicated, to the manuscript.

Regarding the Reviewer's comment on Figure 4a. We would like to highlight that in the original schematic of the workflow, we did include the text "Cell-free operated gene circuit". However, to improve the clarity of the schematic, we have now added the text "CFS" to the corresponding process arrow, as well as the incubation temperatures used for each step.

Regarding the Reviewer's question of "what bottleneck does it specifically address?". As we describe in the abstract of the manuscript, the data we present is focused on demonstrating how the glucose meter interface could provide "a universal reader" for the diagnostic output from gene circuit-based sensors. We agree with the Reviewer's point that the challenges of de-centralizing molecular

diagnostics to the point-of-care are significant and multi-factored in the field. As mentioned above, to avoid confusion for readers we have changed the title to now read “A Glucose Meter Interface for Point-of-Care Gene Circuit-based Diagnostics”.

In response to the comment regarding temperature control and liquid handling. As we describe above, we have incorporated a compact and portable incubator into the manuscript that is capable of providing temperature control for all steps of the protocol (Fig. 5d). Our vision for first adopters of the glucose meter interface is first responders to health crises. A topical example of this could be COVID-19 testing centers in the community, where it may be realistic for trained users to handle samples with pipettes. We have also included the web-based interface for the unambiguous diagnostic interpretation of glucose measurements (Fig. 3d). Of course, as with any technology, the details of how to fully implement applications take time and the practical operationalization of processes are not typically complete at the proof-of-concept stage.

Regarding the estimated cost of diagnostic tests for the glucose meter interface, we have now added these details to the manuscript (pg. 11). Our calculations estimate the cost of the glucose meter-based diagnostic at \$9.26 USD per test using the commercial reagents used in the manuscript (e.g. PURExpress, NASBA).

While the Reviewer correctly notes that the glucose meter interface requires the use of recombinant cell-free systems, which are more costly than cell lysate-based reactions. In recent years protocols for in-house PURE systems generated from engineered microbial consortia have been demonstrated to provide recombinant cell-free reactions for a little as \$0.09 USD per μL ^{1,12}. This would reduce the cost of a gene circuit, glucose meter-based molecular diagnostic even further to an estimated \$3.65/test.

Comment:

7.) For a diagnostic approach, the assay needs to show proper validation, where it is compared against a gold standard with proper statistics, and true positive/negative rates are reported.

Response:

We agree with the Reviewer on the importance of assay validation. We would like to highlight that we do compare the performance of the glucose meter interface to a gold standard RT-qPCR diagnostic¹³ in triplicate and present the data with one-way ANOVA statistical analysis (Fig. 5c). In this work we show the clear discrimination of SARS-CoV-2 genomic RNA from infected patient samples from health patient controls.

The Reviewer also mentions including the calculation of true positive/negative rates for the manuscript. This would require a scale-up of the validation to the level of a patient trial, which we feel is beyond the scope of the manuscript for a number of reasons.

- i. The first is that a patient trial is a separate and subsequent effort that typically follows the report of a new technology development, as it requires significant time and resources. We've actually just received funding to do just this with the technology, along with other practical details raised by the Reviewer as part of a one-year program.
- ii. The challenge of performing a patient trial is multiplied by the current circumstances of reduced facility access during COVID restrictions (40% lab capacity) and, as we discovered with the first patient sample set, SARS-CoV-2 patient samples are difficult to secure.
- iii. As well, the request of a patient trial for the report of a technology is unprecedented and has not been part of other recently published diagnostic technologies from the field of synthetic biology¹⁴

¹⁶. Similarly, the cited work by Du and Zhang did not include patient trials as part of technology validation.

We are certainly headed in the direction of a full patient trial for the technology, but we anticipate that this work will be published separately at a later date.

Comment:

8.) *What is required to go from sample to answer? In particular, what are the sample extraction steps that must be done? The authors need to include description of how realistic sample matrices would be handled.*

Response:

As we discuss above, the challenge of sample preparation at the point-of-care is a long-standing challenge facing the field as a whole. We have added text to the discussion highlighting this challenge (pg. 12) and have included further details on how the glucose meter interface could be used with the different diagnostic inputs in response to comment #4.

Comment:

9.) *The description of how the AmpR and SptR switches is vague, and needs more detail.*

Response:

We thank the Reviewer for highlighting the potential for confusion regarding the function of AmpR and SptR toehold switch-based sensors. We have now added an additional schematic figure in the Supplementary Information that illustrates the function of these sensors (Fig. S2).

Comment:

10.) *The enzymes in Figure 1b should be labeled with Tre, Lac, Phos, to help the reader understand what the enzymes are.*

Response:

We appreciate the Reviewer's suggestion and have now added labels to the schematic in Fig. 1b, along with companion text in the figure legend.

References

1. Lavickova B, Maerkl SJ. A Simple, Robust, and Low-Cost Method To Produce the PURE Cell-Free System. *ACS Synth Biol* [Internet]. 2019;8(2):455–462. Available from: <http://www.ncbi.nlm.nih.gov/pubmed/30632751> PMID: 30632751
2. Massi MN, Shirakawa T, Gotoh A, Bishnu A, Hatta M, Kawabata M. Quantitative detection of *Salmonella enterica* serovar Typhi from blood of suspected typhoid fever patients by real-time PCR. *Int J Med Microbiol. Elsevier GmbH*; 2005 Jun;295(2):117–120. PMID: 15969472
3. Pardee K, Green AA, Ferrante T, Cameron DEE, Daleykeyser A, Yin P, Collins JJ. Paper-based synthetic gene networks. *Cell*. 2014 Nov;159(4):940–954. PMID: 25417167
4. Pardee K, Green AA, Takahashi MK, Braff D, Lambert G, Lee JW, Ferrante T, Ma D, Donghia N, Fan M, Daringer NM, Bosch I, Dudley DM, O'connor DH, Gehrke L, Collins JJ, Collins Correspondence JJ. Rapid, Low-Cost Detection of Zika Virus Using Programmable Biomolecular Components. *Cell*. 2016 May;165(5):1255–1266. PMID: 27160350
5. Myhrvold C, Freije CA, Gootenberg JS, Abudayyeh OO, Metsky HC, Durbin AF, Kellner MJ, Tan AL, Paul LM, Parham LA, Garcia KF, Barnes KG, Chak B, Mondini A, Nogueira ML, Isern S, Michael SF, Lorenzana I, Yozwiak NL, MacInnis BL, Bosch I, Gehrke L, Zhang F, Sabeti PC. Field-deployable viral diagnostics using CRISPR-Cas13. *Science* (80-). 2018;360(6387):444–448. PMID: 29700266
6. Sidstedt M, Rådström P, Hedman J. PCR inhibition in qPCR, dPCR and MPS—mechanisms

and solutions. *Anal Bioanal Chem* [Internet]. 2020 Apr 12;412(9):2009–2023. Available from: <http://link.springer.com/10.1007/s00216-020-02490-2>

7. Pollock N, Westerling J, Sloutsky A. Specimen Dilution Increases the Diagnostic Utility of the Gen-Probe Mycobacterium Tuberculosis Direct Test. *Am J Clin Pathol* [Internet]. 2006 Jul 1;126(1):142–147. Available from: <https://academic.oup.com/ajcp/article-lookup/doi/10.1309/JBQHWC6H4YN6F67Q>
8. Acharya KR, Dhand NK, Whittington RJ, Plain KM. PCR Inhibition of a Quantitative PCR for Detection of Mycobacterium avium Subspecies Paratuberculosis DNA in Feces: Diagnostic Implications and Potential Solutions. *Front Microbiol* [Internet]. 2017;8:115. Available from: <http://www.ncbi.nlm.nih.gov/pubmed/28210245> PMID: 28210245
9. Vázquez-Figueroa E, Chaparro-Riggers J, Bommarius AS. Development of a Thermostable Glucose Dehydrogenase by a Structure-Guided Consensus Concept. *ChemBioChem*. John Wiley & Sons, Ltd; 2007 Dec;8(18):2295–2301.
10. Caurio CFB, Allende OS, Kist R, Vasconcellos ICS, Rozales FP, Reck-Kortmann M, Dalla Lana DF, Alegretti AP, Neto GB, Pasqualotto AC. Cost minimization analysis of an in-house molecular test for cytomegalovirus in relation to a commercial molecular system. *Brazilian J Infect Dis*. Elsevier Editora Ltda; 2020 May;24(3):191–200. PMID: 32450055
11. Latremouille-Viau D, Guerin A, Gagnon-Sanschagrín P, Dea K, Cohen BG, Joseph GJ. Health care resource utilization and costs in patients with chronic myeloid leukemia with better adherence to tyrosine kinase inhibitors and increased molecular monitoring frequency. *J Manag Care Spec Pharm*. Academy of Managed Care Pharmacy (AMCP); 2017 Jan;23(2):214–224. PMID: 28125373
12. Villarreal F, Contreras-Llano LE, Chavez M, Ding Y, Fan J, Pan T, Tan C. Synthetic microbial consortia enable rapid assembly of pure translation machinery. *Nat Chem Biol*. 2018 Jan;14(1):29–35.
13. Ben-Ami R, Klochendler A, Seidel M, Sido T, Gurel-Gurevich O, Yassour M, Meshorer E, Benedek G, Fogel I, Oiknine-Djian E, Gertler A, Rotstein Z, Lavi B, Dor Y, Wolf DG, Salton M, Drier Y. Large-scale implementation of pooled RNA extraction and RT-PCR for SARS-CoV-2 detection. *Clin Microbiol Infect*. Elsevier B.V.; 2020 Sep;26(9):1248–1253. PMID: 32585353
14. Gootenberg JS, Abudayyeh OO, Lee JW, Essletzbichler P, Dy AJ, Joung J, Verdine V, Donghia N, Daringer NM, Freije CA, Myhrvold C, Bhattacharyya RP, Livny J, Regev A, Koonin E V., Hung DT, Sabeti PC, Collins JJ, Zhang F. Nucleic acid detection with CRISPR-Cas13a/C2c2. *Science* [Internet]. 2017 Apr 28;356(6336):438–442. Available from: <http://www.sciencemag.org/lookup/doi/10.1126/science.aam9321> PMID: 28408723
15. Gootenberg JS, Abudayyeh OO, Kellner MJ, Joung J, Collins JJ, Zhang F. Multiplexed and portable nucleic acid detection platform with Cas13, Cas12a, and Csm6. *Science*. 2018 Feb; PMID: 29449508
16. Chen JS, Ma E, Harrington LB, Da Costa M, Tian X, Palefsky JM, Doudna JA. CRISPR-Cas12a target binding unleashes indiscriminate single-stranded DNase activity. *Science*. American Association for the Advancement of Science; 2018 Feb;360(6387):436–439. PMID: 29449511

Reviewers' Comments:

Reviewer #1:

Remarks to the Author:

The authors have addressed my concerns regarding data presentation, and most of my other concerns.

I still have two minor questions for the authors:

1. Regarding glucose clearing - the transport media for many swab types contains sucrose. Are there any concerns that this sucrose will get converted to glucose and interfere with the assay? It would be helpful to address this issue in the paper.

2. The portable incubator appears to be a new addition to the paper. How much does the incubator device cost? It would be helpful if the authors could provide a parts list and a cost breakdown, if possible.

Reviewer #2:

Remarks to the Author:

The authors have sufficiently addressed all of the previously raised comments, and the expanded discussion and details have greatly improved the revised manuscript. The manuscript is suitable for publication.

**Response to Reviewers: Manuscript NCOMMS-20-12233B
“A Glucose Meter Interface for Point-of-Care Gene Circuit-based Diagnostics”**

Dear Dr Pardee,

Your manuscript entitled "A Glucose Meter Interface for Point-of-Care Gene Circuit-based Diagnostics" has now been seen again by our referees, whose comments appear below. In light of their advice I am delighted to say that we are happy, in principle, to publish a suitably revised version in Nature Communications under the open access CC BY license (Creative Commons Attribution 4.0 International License).

We therefore invite you to revise your paper one last time to address the remaining concerns of our reviewers and our editorial requests in the attached document(s). At the same time we ask that you edit your manuscript to comply with our policies and formatting requirements and to maximise the accessibility and therefore the impact of your work.

Please see the attached document(s), listing a number of points that must be addressed. Failure to comply with our editorial requests will cause delays in accepting your manuscript. Please also see the *Nature Communications* formatting instructions for further information.

Reply to Reviewers: NCOMMS-20-12233B

We thank the Reviewers for their enthusiastic, constructive and thoughtful comments throughout the review process, which have led to a significantly improved manuscript. Below, we address additional Reviewers' comments and describe the related updates to the manuscript

REVIEWERS' COMMENTS

Reviewer #1:

Comment:

The authors have addressed my concerns regarding data presentation, and most of my other concerns.

Response:

We thank the Reviewer for their time in reviewing the manuscript and support of the work.

Comment:

I still have two minor questions for the authors:

1. Regarding glucose clearing - the transport media for many swab types contains sucrose. Are there any concerns that this sucrose will get converted to glucose and interfere with the assay? It would be helpful to address this issue in the paper.

Response:

Thank you for raising this potential issue in the implementation of the glucose meter interface. Fortunately, the conversion of sucrose to glucose and fructose subunits requires a

sucrase enzyme (e.g. invertase), which is not present in our system. As a result, if any sucrose was brought into our assay, it would be inert to both the glucose meter test strips and the molecular components. More broadly, if contaminating glucose was present in patient samples or transport medium, we would expect this glucose to be removed during the RNA extraction step. This mode of sample preparation was demonstrated with the SARS-CoV-2 patient samples presented in Fig. 5c. In the manuscript, we discuss glucose mitigation strategies in paragraph one of the Discussion and cite data for each of the three approaches demonstrated.

Comment:

2. The portable incubator appears to be a new addition to the paper. How much does the incubator device cost? It would be helpful if the authors could provide a parts list and a cost breakdown, if possible.

Response:

Thanks for your comment. We were excited to add the portable incubator to the revised manuscript. We have included a Bill of Materials as a supplementary file that includes the parts list, catalog numbers and links to the vendors. We have also included the 3D printer file so that interested readers can build their own devices. The estimated cost of the consumable components is \$92 USD (\$120 CAD) and this has been added to the corresponding methods section.

Reviewer #2:

Comment:

The authors have sufficiently addressed all of the previously raised comments, and the expanded discussion and details have greatly improved the revised manuscript. The manuscript is suitable for publication.

Response:

We appreciate Reviewer's efforts in reviewing our work and their contributions to improving the manuscript.